# Novel Therapeutic Targets on the Horizon: An Analysis of Clinical Trials on Therapies for Bone Metastasis in Prostate Cancer

**DOI:** 10.3390/cancers16030627

**Published:** 2024-01-31

**Authors:** Wenhao Zhou, Wei Zhang, Shi Yan, Kaixuan Zhang, Han Wu, Hongyu Chen, Minfeng Shi, Tie Zhou

**Affiliations:** 1Department of Urology, Shanghai Fourth People’s Hospital, School of Medicine, Tongji University, Shanghai 200434, China; zwh01215@163.com (W.Z.); yanshi1997@163.com (S.Y.); zkx922429@126.com (K.Z.); 15221778695@163.com (H.W.); 2Department of Urology, Changhai Hospital, Naval Medical University, Shanghai 200433, China; zhweismmu@163.com; 3School of Medicine, Tongji University, Shanghai 200092, China; kjchy123@163.com; 4Reproduction Center, Changhai Hospital, Naval Medical University, Shanghai 200433, China

**Keywords:** prostate cancer, bone metastasis, bone-targeted therapies, skeletal-related event

## Abstract

**Simple Summary:**

Prostate cancer is the most common non-cutaneous malignancy among men in the United States. Bone metastases and health are crucial issues in prostate cancer patient management. The aim of our review was to summarize the novel therapeutic targets of prostate cancer with bone metastasis, including treatments to palliate pain and therapies to prevent complications of osseous metastasis. Additionally, our study offers a comprehensive overview of the research on bone metastases in prostate cancer, which can be a valuable resource for researchers in the field.

**Abstract:**

In the absence of early detection and initial treatment, prostate cancer often progresses to an advanced stage, frequently spreading to the bones and significantly impacting patients’ well-being and healthcare resources. Therefore, managing patients with prostate cancer that has spread to the bones often involves using bone-targeted medications like bisphosphonates and denosumab to enhance bone structure and minimize skeletal complications. Additionally, researchers are studying the tumor microenvironment and biomarkers to understand the mechanisms and potential treatment targets for bone metastases in prostate cancer. A literature search was conducted to identify clinical studies from 2013 to 2023 that focused on pain, performance status, or quality of life as primary outcomes. The analysis included details such as patient recruitment, prior palliative therapies, baseline characteristics, follow-up, and outcome reporting. The goal was to highlight the advancements and trends in bone metastasis research in prostate cancer over the past decade, with the aim of developing strategies to prevent and treat bone metastases and improve the quality of life and survival rates for prostate cancer patients.

## 1. Introduction

Prostate cancer (PCa) is a prevalent malignant tumor in the United States, ranking second in terms of mortality rate after lung cancer [1]. There exists a significant disparity in the occurrence rate of prostate cancer between China (10.2/100,000) and North America (73.0/100,000), with both the incidence and mortality rates showing a consistent upward trend in recent years [2,3]. The 2014 China Multicenter Report revealed that a significant proportion of Chinese patients (approximately 30.5%) diagnosed with prostate cancer had already developed distant metastases at the time of initial diagnosis, which is considerably higher compared to the rates observed in North America [4]. Nowadays, the treatment options for patients diagnosed with metastatic prostate cancer (mPCa) have shown significant advancements in recent years. Androgen deprivation therapy (ADT) serves as the primary treatment for this condition. Additional treatment options encompass chemotherapy, new generation hormone therapy, radium-223, and, more recently, radioligand therapy. Special considerations should be directed toward the management of bone health and the prevention of treatment-induced bone loss in these patients [5]. Among individuals diagnosed with castration-resistant prostate cancer (CRPC), bone metastasis is commonly detected in 70% to 90% of patients through imaging examinations [6]. Bone metastases give rise to the occurrence of skeletal-related events (SREs), which encompass severe pain, pathological fracture, spinal cord/intervertebral nerve compression, and hypercalcemia [5]. Preventing and reducing the occurrence of SREs, relieving pain caused by bone metastases, and improving patients’ quality of life are the goals of treatment. The management of bone metastases in prostate cancer has undergone significant advancements due to the enhanced comprehension of the disease’s progression, signaling pathways, mutational characteristics, and mechanisms of drug resistance. In Table 1 and Figure 1, we summarize the main pathways and mechanisms of action of the principal PC therapeutic agents. However, there exists a dearth of data analysis pertaining to drug trials and their progression over the previous decade. Therefore, a systematic review was conducted to examine the development trends of drug therapy for bone metastasis in China and globally from 2013 to 2023. Our review endeavors to draw attention to the biological and clinical significance of bone metastasis, offering a glimpse into potential therapeutic implications in the future.

## 2. Hormonal Therapy

Enzalutamide is a second-generation androgen receptor antagonist and was initially reported in 2009 [7]. In men with metastatic hormone-sensitive prostate cancer (mHSPC), the ARCHES study demonstrated that enzalutamide exhibited a significant reduction in the risk of mortality when compared to placebo (HR: 0.39, 95% CI 0.30–0.50; *p* < 0.001). Additionally, enzalutamide was found to be effective in reducing the occurrence of the first symptomatic skeletal events, castration resistance, and pain progression [8]. A post hoc analysis of the ARCHES study provided additional clarification on the effects of enzalutamide in reducing the risk of radiographic progression of bone metastases (HR: 0.33, 95% CI 0.22–0.49) and the risk of bone metastases with lymph node metastases (HR: 0.31, 95% CI 0.21–0.47) when compared to placebo. Still, there was no significant reduction in the risk of lymph node metastasis [9]. PREVAIL, a double-blind, phase III study [10], met its primary endpoint, radiographic progression-free survival, with a significant advantage in the enzalutamide arm (65% versus 14%, *p* < 0.001). The occurrence of SREs, which was assessed as a secondary endpoint, demonstrated improvement in the enzalutamide group (32% vs. 37%, *p* < 0.001). Additionally, enzalutamide has shown efficacy in patients with low baseline PSA levels (i.e., <10 ng/mL), including those with ≥4 bone metastases and/or visceral disease and <4 bone metastases without visceral disease [11].

Abiraterone, a CYP17 inhibitor that targets a crucial enzyme involved in androgen synthesis, was assessed in the LATITUDE trial [12]. In this phase III trial, a total of 1199 mCSPC patients were randomly assigned in a 1:1 ratio to receive ADT + abiraterone acetate + prednisone versus ADT + dual placebo. Treatment with abiraterone was associated with a statistically significant survival advantage (not reached vs. 34.7 months), and the median length of radiographic progression-free survival was 33.0 months in the abiraterone group and 14.8 months in the placebo group. A post hoc analysis revealed that abiraterone acetate led to improvements in bone pain, fatigue symptoms, and overall health-related quality of life. Patients in the abiraterone group had a longer median time to worst pain intensity, worst fatigue intensity, and functional deterioration condition [13]. The STAMPEDE trial examined the efficacy of abiraterone acetate in combination with prednisolone and ADT versus ADT alone in patients with locally advanced or metastatic PC. After 3 years of treatment, survival improved (83% versus 76%, HR 0.63; *p* < 0.001) and the risk of SREs decreased (12% versus 22%, HR 0.46, *p* < 0.001) in the combination group [14]. Abiraterone was assessed in the COU-AA-302 trial, which examined the efficacy of abiraterone acetate in combination with prednisone compared to placebo and prednisone in mCRPC patients who had not previously received chemotherapy. The pain progression was 26.7 months in the abiraterone group and 18.4 months in the prednisone group (HR, 0.82; 95% CI, 0.67 to 1.00; *p* < 0.05), and the advantage in radiographic progression-free survival reached statistical significance (16.5 months versus 8.3 months; *p* < 0.001) [15].

Apalutamide was first described in 2012 as a novel antiandrogen for prostate cancer [16]. The TITAN trial evaluated the therapeutic efficacy of adding oral apalutamide to ADT for the treatment of adult men with metastatic castration-sensitive prostate cancer (mCSPC). The 24-month OS rates were 82.4% in apalutamide plus ADT recipients and 73.5% in placebo plus ADT recipients (*p* = 0.005). There was no substantial difference between the two groups in the time to the skeletal-related events of prostate cancer [16]. The primary objective of the TITAN study was to assess the therapeutic effectiveness of combining oral apalutamide with ADT (not reached, HR 0.80) [17].

Approximately 8% of CRPC patients have the androgen receptor splice variant-7 (AR-V7) blood biomarker [18], which has been linked to resistance against enzalutamide and abiraterone [19]. Galeterone has been reported to inhibit AR signaling through multiple mechanisms: CYP17 inhibition, AR competitive antagonism, and induction of AR and AR-V7 protein degradation [20]. In a randomized phase III trial [21], CRPC patients with AR-V7 expression were randomly assigned to receive either galeterone or enzalutamide in an open-label manner. However, this trial did not achieve its primary objective due to a high number of patients discontinuing the study as a result of disease progression. Therefore, there is an urgent need for alternative treatments for circulating tumor cells expressing AR-V7 in mCRPC.

Bipolar androgen therapy (BAT) is an emerging treatment option for patients with CRPC. BAT has shown promise in restoring drug sensitivity in some patients, particularly to medications like Abiraterone and Enzalutamide. This therapy has demonstrated significant advantages in the treatment of CRPC patients [22]. The RESTORE trial was a single-arm, multicohort study, focused on CRPC patients. The results from this trial indicated that patients who had previously experienced progression on enzalutamide showed a 52% PSA50 response rate to enzalutamide after undergoing BAT. Similarly, patients who had previously progressed on abiraterone demonstrated a 16% PSA50 response to abiraterone after BAT. However, this study did not investigate the effects of BAT on SREs [23,24].

More than 50% of prostate cancer patients receiving long-term ADT have significant associated metabolic consequences, such as insulin resistance and metabolic syndrome [25,26]. A phase II trial observed that the use of high-dose metformin in mCRPC reduced PSA secretion and delayed the progression of prostate cancer [27].A cohort study based on a prostate cancer population showed that patients with hyperlipidemia may have prolonged survival with metformin and statins while undergoing radiotherapy [28]. Ongoing clinical trials will help elucidate the role of metformin in the treatment of locally advanced or metastatic prostate cancer [29,30].

## 3. Chemotherapy

In 2004, the United States Food and Drug Administration (FDA) approved the use of Docetaxel, a taxane drug that binds to tubulin, for the treatment of mCRPC [31]. Another taxane drug called cabazitaxel was also approved by the FDA in 2010 as a second-line salvage chemotherapy for prostate cancer [32]. A phase II trial found that a weekly treatment schedule of 10 mg/m^2^ of docetaxel resulted in a 34.9% prostate-specific antigen (PSA) response rate, with lower toxicity rates of 14.2% neutropenia and 35.7% diarrhea [33]. However, the GETUG-AFU 15 trial suggested that the addition of docetaxel to ADT should not be used as a first-line treatment for mCSPC as it did not improve overall survival [34]. On the other hand, the CHAARTED trial showed that adding docetaxel to ADT in early-stage prostate cancer improved overall survival, particularly in high-volume disease, but did not show a clear survival benefit in low-volume disease [35,36]. One potential reason for the discrepancy between the GETUG-AFU 15 and CHAARTED trials is the lack of statistical power in the former. The STAMPEDE trial reported a survival benefit with upfront docetaxel in patients with mCSPC, regardless of metastatic burden [37]. According to the National Comprehensive Cancer Network (NCCN) guidelines, patients with high-volume metastatic disease who are suitable for chemotherapy should receive ADT in combination with docetaxel, along with either abiraterone or darolutamide. The ARASENS trial found that adding darolutamide to ADT and docetaxel improved the overall survival of mHSPC patients with a similar rate of side effects compared to using a placebo with ADT and docetaxel [38]. The PEACE-1 trial demonstrated that using abiraterone in combination with ADT improved overall survival and progression-free survival in patients with de novo mCSPC, with only slight increases in treatment-related side effects [39]. Additionally, the findings from the ENZAMET trial suggested that adding enzalutamide should be considered for patients with mCSPC who are treated with docetaxel [40,41]. While studies have shown the benefits of doublet therapy with ADT plus androgen receptor signaling inhibitors (ARSIs), as well as the benefits of triplet therapy with ADT plus docetaxel and ARSIs, a direct comparison between doublet therapy and triplet therapy for mCSPC has not been conducted [38,39]. However, patients with low-volume disease appear to have increased treatment benefit from ARSI doublet therapy compared to docetaxel and ADT [42]. It is noteworthy that the overall survival (OS) rate is comparatively lower in African American individuals than in Caucasian individuals among patients diagnosed with prostate cancer. However, after administration of docetaxel, the OS rate in African American patients approached parity with that of Caucasian patients. This phenomenon may be attributed to racial disparities in drug responsiveness. The documented benefits of docetaxel or cabazitaxel in terms of OS are well established. However, there is currently no conclusive evidence regarding their impact on pain management and the potential delay or prevention of SREs in patients with mCRPC [43].

Table 2 provides an overview of the phase III trials on prostate cancer and their outcomes.

## 4. Bone-Modifying Agents

Osteoporosis is commonly observed in patients with prostate cancer. Studies have shown that a significant percentage of hormone-naïve PC patients (ranging from 3.9% to 37.8%) develop osteoporosis even before receiving any oncological treatment. This suggests that PC itself may be a risk factor for the loss of bone mineral density (BMD) due to its promotion of bone resorption [46]. ADT is designed to reduce testosterone by up to 95% and lower estrogen, but it also causes an increase in bone resorption to altering the balance between osteoblasts and osteoclasts and results in a rapid decline in BMD. The decline in BMD begins shortly after the initiation of ADT and continues throughout the treatment period [47]. The duration of ADT is directly proportional to the risk of osteoporotic fractures [48].

The efficacy of bone health agents, such as zoledronic acid and denosumab, in reducing the occurrence of SREs and delaying their onset in patients with bone metastases from prostate cancer has been extensively studied. The NCCN guidelines for the treatment of osteoporosis in prostate cancer patients receiving ADT recommend several strategies. They suggest calcium and vitamin D3 supplementation as a standard approach. Additionally, for men aged 50 years and older who have low bone mass in the femoral neck (with T values falling between −1.0 and −2.5), the NCCN advises considering further therapy options such as denosumab or zoledronic acid. Zoledronic acid is the most commonly used bisphosphonate for managing bone metastasis in prostate cancer patients due to its reported ability to prolong the time to SREs and alleviate bone pain [49]. Despite having similar rates of overall survival and SREs, zoledronic acid demonstrated superior efficacy in managing pain compared to clodronate [50] (Appendix A). However, the effectiveness of zoledronic acid varies among studies, and some have yielded inconclusive results. For instance, a phase III clinical trial demonstrated that patients with mCSPC and bone metastases treated with zoledronic acid and ADT experienced a significantly shorter time to the first SRE (18.8 months) compared to those treated with ADT alone [51]. Conversely, the ALLIANCE 90202 trial found no association between zoledronic acid use and a reduced risk of SREs in men with mCSPC [52]. In TROG 03.04 RADAR trail [53], 18 months of androgen suppression plus radiotherapy is a more effective regimen for treating locally advanced prostate cancer, but the addition of zoledronic acid to this regimen does not significantly improve OS. Similarly, the TRAPEZE study reported that zoledronic acid did not prolong OS [54]. Moreover, in patients at high risk for localized PCa, zoledronic acid proved to be ineffective in preventing bone metastases [55]. Zoledronic acid has been shown to improve BMD when administered at various dosing intervals. In the United States, the approved use of zoledronic acid specifies that it should be used when prostate cancer has progressed despite hormone therapy. For patients with mCRPC and skeletal metastases, zoledronic acid has been utilized in accordance with the EAU guidelines to mitigate the occurrence of SREs [56]. The currently approved dose in most clinical trials is 4 mg intravenously every 3–4 weeks [57,58,59].

Numerous trials have examined the effectiveness of zoledronic acid in preventing BMD decline, but none of these trials were designed to detect a difference in fracture risk [60]. Denosumab, on the other hand, is a fully humanized monoclonal antibody that binds to and neutralizes RANKL, a protein involved in bone resorption. By inhibiting signaling through its target RANK, denosumab suppresses bone resorption by osteoclasts [61]. A post hoc analysis of three phase III trials compared denosumab to zoledronic acid in terms of reducing the risk of SREs, including both first-time and subsequent events [62]. The analysis found that denosumab was more effective than zoledronic acid in preventing SREs, regardless of factors such as Eastern Cooperative Oncology Group performance status, location and number of bone metastases, presence or absence of visceral metastases, and urinary N-telopeptide level. The standard dosage for denosumab is 120 mg administered subcutaneously every 4 weeks and there is evidence to suggest that administering bone-modifying agents every 12 weeks instead of every 4 weeks may be equally effective in preventing SREs [63,64]. Thus, prolonging the interval between doses of bone-modifying agents may help avoid the risk of adverse events such as osteonecrosis of the jaw (ONJ) without compromising SRE prevention.

A phase I clinical trial was conducted to evaluate the efficacy and safety of a novel bone targeting polybisphosphonate called OsteoDex with bifunctional cytotoxic properties [65]. The findings of the trial demonstrated that OsteoDex was well tolerated, resulting in minimal adverse effects, and exhibited a notable therapeutic effect, particularly in the highest dose group. These results highlight the potential of developing targeted therapies that specifically address the underlying mechanisms of bone metastasis in prostate cancer. Furthermore, combining different treatment modalities in a comprehensive treatment approach shows promise as a strategy to enhance therapeutic outcomes in this context.

## 5. Radionuclide Therapy

Radium-223 (Ra-223) is a radioactive isotope that emits α particles and has a physical half-life of 11.43 days. It has been observed that Ra-223 can cause DNA double-strand breaks not only in cancer cells but also in osteoblasts and osteoclasts [66]. In the phase III ALSYMPCA trial [67], Ra-223 was found to significantly improve the overall survival of mCRPC patients with bone metastases compared to placebo (14.0 months vs. 11.2 months, HR 0.70, *p* < 0.001). It also improved the quality of life, reduced the incidence of myelosuppression, and delayed the occurrence of skeletal-related events (15.6 months vs. 9.8 months, HR 0.66, *p* < 0.001). A subgroup analysis of the ALSYMPCA trial showed that the survival benefit of Ra-223 was consistent regardless of prior docetaxel use [68]. Another phase III trial reported that Ra-223 led to a significant improvement in mCRPC patients’ quality of life and an increase in overall survival by 3.6 months [69]. Hijab A. et al. found that patients with mCRPC, particularly those treated with Ra-223, are at risk of fractures [70]. The ERA-223 trial [71], which included 806 patients with prostate cancer, showed that the combination of abiraterone and Ra-223 did not delay skeletal-related events in patients with mCRPC and may actually increase the incidence of fractures. Therefore, the use of Ra-223 plus abiraterone is not recommended for the treatment of mCRPC. Men with mCRPC should receive bone-modifying agents to reduce their risk of fragility fractures. Enzalutamide is also being evaluated in combination with Ra-223 in a phase II trial. This combination has shown potential in decreasing bone metabolic markers, improving outcomes, and prolonging overall survival, radiographic progression-free survival, and time to the next treatment [72,73]. In a phase II, open-label, single-arm, multicenter study, Ra-223 was found to be safe regardless of concurrent use of androgen signaling inhibitors. Furthermore, patients who received three or fewer anticancer therapies had a longer survival with Ra-223 [74].

PSMA, a transmembrane glutamate carboxypeptidase, is expressed in over 90% of metastatic prostate cancer lesions and its expression increases with higher Gleason scores [75,76]. In a retrospective study involving 10 patients with mCRPC, treatment with ^177^Lu-PSMA-617 resulted in a decrease in prostate-specific antigen (PSA) levels in 7 patients, with 5 of them experiencing a PSA decrease of more than 50% after 8 weeks [77]. Another study enrolled 52 mCRPC patients who received 3–6 cycles of ^177^Lu-PSMA-617, in which, 30 patients (44.2%) experienced a PSA response, and the median overall survival (OS) for all patients was 60 weeks [78]. In a prospective single-arm phase II trial, 30 men with mCRPC were administered intravenous injections of ^177^Lu-PSMA-617 [79]. Seventeen patients (57%) experienced PSA responses, and most patients showed improved toxic effects and pain during treatment, indicating the antitumor activity of ^177^Lu-PSMA-617. A pilot study also reported that ^177^Lu-PSMA-617 improved quality of life by increasing global health and mitigating disease-related pain [80]. The FDA approved ^177^Lu-PSMA-617 as a therapeutic option in pretreated mCRPC patients in March 2022, based on the results of the most advanced phase III VISION trial [81]. The phase III VISION trial evaluated ^177^Lu-PSMA-617 in 831 patients with mCRPC [82]. When compared to standard of care (SOC) alone, ^177^Lu-PSMA-617 plus SOC significantly prolonged rPFS (median, 8.7 vs. 3.4 months; *p* < 0.001) and median OS (15.3 vs. 11.3 months; *p* < 0.001). The results of the TheraP trial compared ^177^Lu-PSMA-617 to cabazitaxel in 200 men with mCRPC [83]. In this phase II trial, ^177^Lu-PSMA-617 was shown to lead to a higher PSA response (65 vs. 37%, *p* < 0.0001) along with fewer grade 3 or 4 adverse events (33% vs. 53%). Collectively, these findings demonstrate that ^177^Lu- PSMA-617 is an effective therapy for patients with mCRPC, and can decrease the disease-related pain. However, its role in alleviating bone pain is not yet defined.

MEDI3726 is an antibody–drug conjugate that exhibits high potency. It is composed of an engineered version of the anti-PSMA IgG1k antibody (J591) that is specifically conjugated with pyrrolobenzodiazepine dimers at a drug–antibody ratio of approximately 2 [84]. A phase I study was conducted, enrolling 33 patients with mCRPC who had previously failed abiraterone, enzalutamide, or a taxane-based therapy [85]. Among these patients, treatment-related adverse events were observed in 30 individuals (90.9%), leading to discontinuation of treatment in 11 patients (33.3%). Unfortunately, the results of this study were insufficient to establish the safety and efficacy of MEDI3726. As of now, no further clinical trials have been conducted with MEDI3726.

## 6. Radiation Therapy

Conventional external beam radiation therapy (cEBRT) is commonly used as the primary treatment for painful spine metastases. Previous studies, conducted retrospectively or with single-arm designs, have demonstrated the effectiveness of cEBRT in terms of delaying disease progression and initiating ADT [86,87]. Hoskin P et al. reported single radiation therapy for metastatic prostate bone pain is similar to a single infusion of ibandronate [88]. However, the HORRAD trial revealed that additional radiotherapy did not result in improved overall survival for patients with bone metastatic prostate cancer [89]. In recent years, there have been significant advancements in radiotherapy technologies, one of which is stereotactic body radiotherapy (SBRT). SBRT is a newly introduced approach that has shown promise in enhancing treatment efficacy while minimizing treatment-related adverse events [90].

A phase II study examined the effectiveness of SBRT for one to three recurrent metastatic lesions in asymptomatic PCa patients treated with radical prostatectomy, primary radiotherapy, or a combination of both. After a median follow-up period of 3 years, patients who received metastasis-directed therapy exhibited superior ADT-free survival compared to those who underwent surveillance (21 vs. 13 months) [91]. After a 5-year period, the rates of survival of oligometastatic prostate cancer patients without androgen deprivation therapy were 34% for the metastasis-directed therapy group and 8% for the surveillance group [92]. The ORIOLE study included 54 participants with recurrent oligometastatic hormone-sensitive prostate cancer [93] and this prospective phase II RCT revealed that after 6 months of follow-up, the intervention arm demonstrated higher PFS, with only 19% of patients experiencing progression compared to 61% in the control group (median progression-free survival: not reached vs. 5.8 months; hazard ratio: 0.30; *p* = 0.002). However, the RTOG0631 trial did not meet its primary endpoint of demonstrating the superiority of SBRT for pain response at 3 months [94]. Recently, lots of studies involving SBRT for the treatment of mCRPC had reported that SBRT significantly prolonged the time to symptomatic progression. However, in these studies, the target of SBRT also included sites other than bone metastasis sites. Thus, the role of SBRT in treating bone metastasis in CRPC patients is not clear.

## 7. Immunotherapy

Immunotherapy has demonstrated limited effectiveness in treating metastatic prostate cancer compared to its success in other types of cancer such as melanoma or renal cell carcinoma. Various studies have investigated the potential of immune checkpoint inhibitors (ICIs) as standalone treatments for prostate cancer, but unfortunately, no favorable outcomes have been observed [95]. Ipilimumab as a single-agent therapy was evaluated in two extensive phase III trials, and did not demonstrate a significant improvement in overall survival (OS) in either study [96,97]. However, ipilimumab has demonstrated the ability to extend the median overall survival in a specific subgroup of patients with mCRPC who do not have visceral disease and have favorable laboratory values [96]. The use of combination therapies shows more potential and offers a reason for optimism. The combination of nivolumab and rucaparib has shown activity in men with mCRPC and bone metastases who have undergone chemotherapy or are chemotherapy-naive, especially in those with BRCA1/2 mutations [98]. Further research is required to determine whether the addition of nivolumab is associated with increased effectiveness when combined with rucaparib. In a separate phase II trial, the efficacy of nivolumab and docetaxel was examined in chemo-naïve patients with mCRPC and bone metastases who were already receiving androgen deprivation therapy (ADT). The objective response rate (ORR) in patients with measurable disease was found to be 36.8%, while the PSA response rate was 46.3%. It is believed that immunotherapy treatment may enhance the effects of docetaxel [99]. Another study investigating the combination of atezolizumab and Ra-223 in mCRPC patients, with bone and lymph node and/or visceral metastases, did not demonstrate any clear clinical benefits [100]. Furthermore, several trials have been conducted to evaluate the efficacy of dual immune checkpoint inhibitor therapies, including the Phase II CheckMate 650 trial, which examined the effectiveness of ipilimumab and nivolumab in mCRPC (78/90 with bone metastasis) patients previously treated with docetaxel [101]. In this trial, a number of patients experienced a reduction in both tumor size and PSA levels (ranging from 75% to 100%) in the cohorts receiving the ipilimumab and nivolumab combination [102].

Olaparib, a PARP inhibitor, has demonstrated significant efficacy in patients with mCRPC who have BRCA1 and BRCA2 mutations, with a response rate of 88%. The TOPARP-A multicenter phase II clinical study revealed that in mCRPC patients who had previously received treatment with docetaxel, cabazitaxel, abiraterone acetate, and enzalutamide, those with BRCA and ATM mutations experienced a longer median radiologic progression-free survival compared to those without mutations after receiving olaparib treatment (9.8 months vs. 2.7 months) [103]. Another study, known as Cohort A of the KEYNOTE 365 study [104], investigated the combination of pembrolizumab and olaparib in 102 mCRPC patients (24/102 with bone metastasis) who had previously received docetaxel treatment. The study reported an objective response rate of 8.5%, a radiologic progression-free survival of 4.5 months, and a median overall survival of 14 months. These findings suggest that patients with DNA damage repair mutations may derive additional benefits from the combination of immunotherapy and PARP inhibitors. The simultaneous use of a PARP inhibitor and a tyrosine kinase inhibitor (TKI) has been recently evaluated in men with mCRPC [105]. The combination of cediranib and olaparib demonstrated a significant improvement in rPFS compared to olaparib alone, as indicated by a hazard ratio (HR) of 0.617 (95% CI, 0.392 to 0.969; *p* = 0.0359). However, it is important to note that in these studies of immune therapy for mCRPC with bone metastasis, there are no specific data on SREs.

PCa is an ideal target for cancer vaccines [106]. Sipuleucel-T, a treatment commonly used in the management of prostate cancer, has shown the greatest efficacy in patients with a lower disease burden who also received the vaccine [107]. However, a phase Ib trial investigating the combination of sipuleucel-T with atezolizumab in patients with mCRPC revealed that only 4.3% of participants experienced an objective response [108]. To establish the true potential of this combination therapy, further investigations involving larger cohorts are necessary.

In a phase I trial conducted by Narayan et al. [109], CAR-T cells were employed as a treatment for mCRPC. The study revealed a significant reduction of over 98% in PSA levels following this therapy. Furthermore, it was observed that only five out of the thirteen patients experienced grade ≥ 2 cytokine release syndrome, indicating a promising and encouraging outcome.

The present methods employed for the clinical treatment of patients with mPCa are depicted in Figure 2. For mCRPC patients with bone metastasis, similar to the EAU guidelines, bone-modifying agents have been strongly recommended.

## 8. Conclusions

It is of utmost importance to raise awareness within both the oncology and medical communities regarding the significance of maintaining bone health before and during prostate cancer treatments. In addition to bone-modifying agents, recent advancements have introduced therapies such as Ra-223 and ARSIs, which have demonstrated the potential to prevent SREs and enhance the quality of life for patients. Furthermore, immune therapy has shown promising outcomes for managing bone metastasis in patients with mCRPC, even though its precise role in preventing SREs remains undefined. Looking ahead, the prospect of combination therapy involving PARP inhibitors, tyrosine kinase inhibitors (TKIs), or ^177^Lu-PSMA-617 offers new avenues for the management of mCRPC patients with bone metastases, potentially ushering in innovative approaches to treatment.

## Figures and Tables

**Figure 1 cancers-16-00627-f001:**
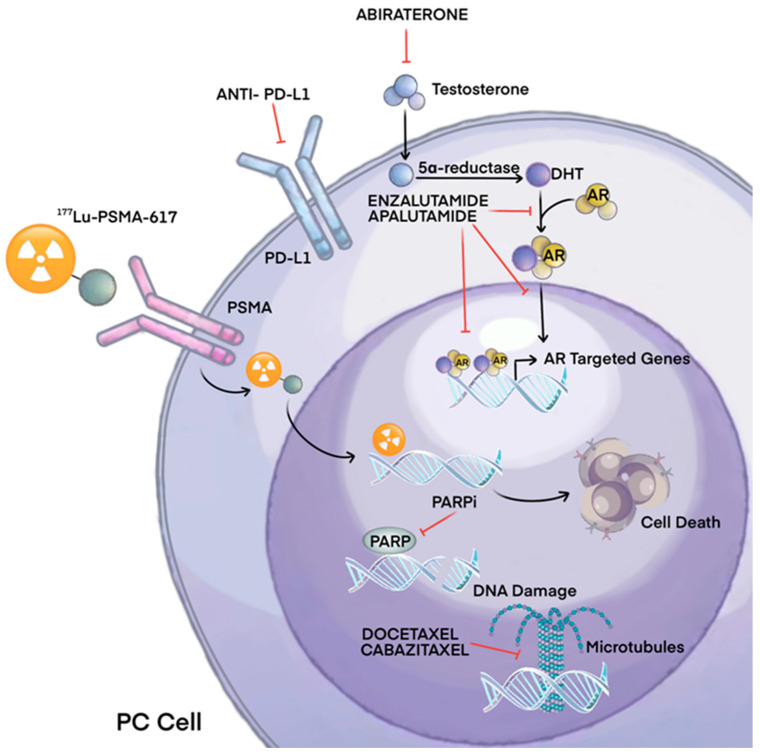
Main mechanisms of therapeutic agents for prostate cancer.

**Figure 2 cancers-16-00627-f002:**
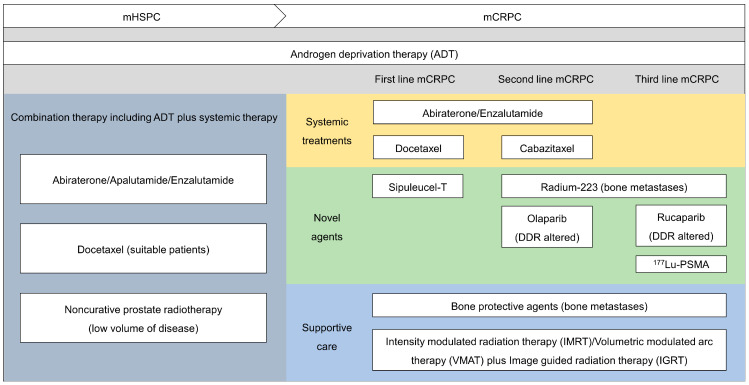
Clinical management options for patients diagnosed with mPCa.

**Table 1 cancers-16-00627-t001:** Drugs and their mechanisms of actions against prostate cancer.

Drug	Action	Mechanism
Abiraterone	Inhibition of androgen synthesis	Inhibits CYP17, reduces androgen production
Enzalutamide	Antagonization of androgen action	Androgen receptor inhibitor, blocks testosterone effects
Bicalutamide		Blockade of the AR
Apalutamide		Prevent AR translocation, DNA binding, and AR–mediated transcription
Docetaxel Cabazitaxel	Inhibition of mitosis	Tubulin inhibition
Radium-223	Alpha radiation, gamma rays	Targets bone metastases, emits alpha particles
^177^Lu-PSMA-617 MEDI3726	Inhibition of growth signals	Binding and internalization of the PSMA ligands triggers cell death
Ipilimumab	Checkpoint (CTLA-4) inhibitor	Increases antitumor T-cell responses
Olaparib Rucaparib	PARP inhibitor	Inhibition of DNA repair
Pembrolizumab	PD-1 inhibitor	Regulates T cell activation
Sipuleucel-T	Immunotherapy	Autologous vaccine
CAR-T		Targeted PSMA

**Table 2 cancers-16-00627-t002:** Comparison of clinical trials investigating novel treatments in mPCa.

Study	Setting	Therapeutic Agent	N	Patient Population	OS	Time to First SRE	Median Follow-Up (Months)
PREVAIL [10]	mCRPC	Enzalutamide vs. placebo	1717 (872 vs. 845)	White (669 vs. 655) Asian (85 vs. 82)	32.4. mo vs. 30.2 mo (*p* < 0.001)	median 31.1 mo vs. 31.3 mo, (*p* < 0.001)	N/A
ARCHES [9]	mHSPC	Ezalutamide + ADT vs. placebo + ADT	1150 (574 vs. 576)	White (466 vs. 460) Asian (75 vs. 80)	NR	NR	14.4
LATITUDE [13]	mCSPC	Abiraterone + prednisone + ADT vs. placebo + ADT	1199 (597 vs. 602)	N/A	NR vs. 34.7 mo (*p* < 0.001)	NR vs. NR (*p* = 0.009)	30.4
STAMPEDE [15]	mHSPC	Abiraterone + prednisone + ADT vs. placebo + ADT	1917 (960 vs. 957)	N/A	83% vs. 76% at 3 y (*p* < 0.001)	88% vs. 78% at 3 y (*p* < 0.001)	40.0
TITAN [17]	mHSPC	Apalutamide + ADT vs. placebo + ADT	1052 (525 vs. 527)	White (354 vs. 365) Asian (119 vs. 110)	NR (82.4% vs. 73.5% at 24 mo, *p* = 0.005)	NR	22.7
CHAARTED [36]	mHSPC	Docetaxel + ADT vs. ADT	790 (397 vs. 393)	White (344 vs. 330) Black (39 vs. 37)	57.6 mo vs. 44.0 mo (*p* < 0.001)	No data	28.9
CARD [44]	mCRPC	Cabazitaxel vs. ARSI	255 (129 vs. 126)	N/A	13.6 mo vs. 11.0 mo (*p* = 0.008)	NR vs. 16.7 mo	9.2
FIRSTANA [45]	mCRPC	Cabazitaxel 20 mg/m^2^ vs. 25 mg/m^2^ vs. docetaxel 75 mg/m^2^	1168 (389 vs. 388 vs. 391)	White (365 vs. 360 vs. 363) Asian (13 vs. 17 vs. 17)	24.5 mo vs. 25.2 mo vs. 24.3 mo	No data	N/A
GETUG-AFU 15 [34]	mCSPC	Docetaxel + ADT vs. ADT	385 (192 vs. 193)	N/A	58.9 mo vs. 54.2 mo (*p* = 0.955)	No data	50.0
ARASENS [38]	mCSPC	Darolutamide + ADT + docetaxel vs. placebo + ADT	1306 (651 vs. 655)	White (345 vs. 333) Asian (230 vs. 245)	NR vs. 48.9 mo (*p* < 0.001)	NR vs. NR (*p* = 0.02)	43.7
ENZAMET [40]	mHSPC	Ezalutamide + ADT vs. ADT	1125 (562 vs. 563)	N/A	NR (80% vs. 72% at 36 mo) (*p* = 0.002)	No data	34.0
PEACE-1 [39]	mCSPC	ADT vs. ADT + radiotherapy vs. ADT + abiraterone vs. ADT + radiotherapy + abiraterone	1173 (296 vs. 293 vs. 292 vs. 291)	N/A	4.46 y vs. 2.03 y (with or without abiraterone, *p* < 0.001)	No data	52.8

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
