# Peer review of "Novel Therapeutic Targets on the Horizon: An Analysis of Clinical Trials on Therapies for Bone Metastasis in Prostate Cancer"

_cancers, 2024, doi:10.3390/cancers16030627_

Round 1

Reviewer 1 Report

Comments and Suggestions for Authors

This is a comprehensive overview about the different treatment modalities for advanced prostate cancer (ie. with bone metastases).

I have some comments:

1. The role of Zoledronic acid should be more specific rather than describing the existing studies: What are the recommendations of EAU-guidelines?

2. The Conclusions are not precise:

What should be the treatment algorithm in 2023?

What recommend The EAU-guidelines

After this nice review, the authors should provide us with actual recommendations!

3. There is a spelling error : In a phase I trial conducted by Narayan et al[99]

Author Response

Paper ID: cancers-2714324

Paper Title: Novel therapeutic targets on the horizon: An analysis of clinical trials on therapies for bone metastasis in prostate cancer

We feel great thanks for your professional review work on our article. As you are concerned, there are several problems that need to be addressed. According to your nice suggestions, we have made extensive corrections to our previous draft, the detailed corrections are listed below.

  1. The role of Zoledronic acid should be more specific rather than describing the existing studies: What are the recommendations of EAU-guidelines?

Response: Thank you for your suggestion. The added information is as follows (see details in lines 216-218 of the revised manuscript).

​

  1. The Conclusions are not precise:

What should be the treatment algorithm in 2023?

What recommend The EAU-guidelines

After this nice review, the authors should provide us with actual recommendations!

Response: Thank you for your suggestion, according to your suggestion, we rewrote the conclusion. At the same time, we also add Figure2 in the revised manuscript to comment the recommendation from EAU-guidelines

  1. There is a spelling error : In a phase I trial conducted by Narayan et al[99]

Response: Thanks for your help. We feel really sorry for our carelessness. We have corrected it in the revised manuscript.

Reviewer 2 Report

Comments and Suggestions for Authors

This is a review of therapeutic options for metastatic prostate cancer, which has a clinical significance. However, this review can be improved to be followed by clinical and the basic science researchers in a better way.

1.      The author titled the review as the “Novel therapeutic targets” for prostate cancer. But they enlisted the existing therapies and the trials. This can be improved if they mention the molecular mechanisms of action of the drugs/ therapies used in the trials. In some cases, they mentioned it. A tabular form of this drugs/ therapies and their mechanisms of actions against prostate cancer will be helpful to comprehend.

2.      The clinical trial mentioned here are important. If the author can include other clinical status of the patient cohort, such as diabetic status, cholesterol levels and statin and anti-diabetic drug consumption with the prostate cancer therapies. The statin/ metformin intake, diabetic condition interfere the therapies. Therefore, including these clinical conditions will improve the review.

3.      The use of abbreviations sometimes interfering the flow of the reading of the reader. Reducing use of abbreviations will help to improve the reading flow of the readers.

4.      Therapy response differ according to the genetic makeup and difference in population, worldwide. Therefore, the mentioning of population status in the clinical trials will also help the reader to understand the effect of the therapies in different populations.

5.      The phase II /III trials, those failed may also worth mentioning briefly because the reason of failing will help to understand better designing the future trials. A brief tabular form of these trials can be included. 

Comments on the Quality of English Language

Minor editing of English language required

Author Response

Paper ID: cancers-2714324

Paper Title: Novel therapeutic targets on the horizon: An analysis of clinical trials on therapies for bone metastasis in prostate cancer

We feel great thanks for your professional review work on our article. As you are concerned, there are several problems that need to be addressed. According to your nice suggestions, we have made extensive corrections to our previous draft. We have added necessary data to supplement our results and edited our article extensively. The detailed corrections are listed below.

  1. The author titled the review as the “Novel therapeutic targets” for prostate cancer. But they enlisted the existing therapies and the trials. This can be improved if they mention the molecular mechanisms of action of the drugs/ therapies used in the trials. In some cases, they mentioned it. A tabular form of this drugs/ therapies and their mechanisms of actions against prostate cancer will be helpful to comprehend.

Response: Thank you for your suggestion, according to your suggestion, we added Table1 in the revised manuscript.

​

  1. The clinical trial mentioned here are important. If the author can include other clinical status of the patient cohort, such as diabetic status, cholesterol levels and statin and anti-diabetic drug consumption with the prostate cancer therapies. The statin/ metformin intake, diabetic condition interfere the therapies. Therefore, including these clinical conditions will improve the review.

Response: Thank you for your valuable suggestion and we agree with your opinion greatly. We again reviewed the literature referenced in the manuscript about the clinical status of the patient cohort, particularly regarding diabetic status, cholesterol levels, and the consumption of statins and anti-diabetic drugs. However, we were unable to access detailed demographic or clinical status data that would provide the specific information mentioned by the reviewer. But we think the reviewer’s suggestion is very important and in the near future we can further explore the role of diabetic status, cholesterol levels, and the consumption of statins and anti-diabetic drugs in therapies on prostate cancer.

Otherwise, we searched for the role of metformin and statins on prostate cancer therapy, which we described in the revised manuscript (see details in lines 133-140)

  1. The use of abbreviations sometimes interfering the flow of the reading of the reader. Reducing use of abbreviations will help to improve the reading flow of the readers.

Response: Thanks for your nice suggestions. In the revised manuscript, certain abbreviations have been removed.

  1. Therapy response differ according to the genetic makeup and difference in population, worldwide. Therefore, the mentioning of population status in the clinical trials will also help the reader to understand the effect of the therapies in different populations.

Response: We sincerely appreciate the valuable comments. Data on racial disparities in men with mPCa are limited. As per the revised manuscript, the term "patient population" has been included in Table 2. We have subsequently incorporated additional components into the revised manuscript. (see details in lines 171-174)

  1. The phase II /III trials, those failed may also worth mentioning briefly because the reason of failing will help to understand better designing the future trials. A brief tabular form of these trials can be included. 

Response: Thank you for your suggestion, according to your suggestion, we added TableS1 in the revised manuscript.

  1. Minor editing of English language required

Response: We invited an English-speaking native to edit the language in the manuscript and hope that the correction will meet with approval.
